

# Learning structural bioinformatics and evolution with a snake puzzle

Gonzalo S. Nido[1,2,*], Ludovica Bachschmid-Romano[3,*], Ugo Bastolla[4] and Alberto Pascual-García[4,5]

[1] Department of Neurology, Bergen University, Bergen, Norway
[2] Department of Clinical Medicine, Bergen University, Bergen, Norway
[3] Department of Artificial Inteligence, Technische Universität Berlin, Berlin, Germany
[4] Centro de Biología Molecular "Severo Ochoa," Universidad Autónoma de Madrid, Madrid, Spain
[5] Department of Life Sciences, Imperial College London, Ascot, Berkshire, United Kingdom
[*] These authors contributed equally to this work.

## ABSTRACT

We propose here a working unit for teaching basic concepts of structural bioinformatics and evolution through the example of a wooden snake puzzle, strikingly similar to toy models widely used in the literature of protein folding. In our experience, developed at a Master's course at the Universidad Autónoma de Madrid (Spain), the concreteness of this example helps to overcome difficulties caused by the interdisciplinary nature of this field and its high level of abstraction, in particular for students coming from traditional disciplines. The puzzle will allow us discussing a simple algorithm for finding folded solutions, through which we will introduce the concept of the configuration space and the contact matrix representation. This is a central tool for comparing protein structures, for studying simple models of protein energetics, and even for a qualitative discussion of folding kinetics, through the concept of the Contact Order. It also allows a simple representation of misfolded conformations and their free energy. These concepts will motivate evolutionary questions, which we will address by simulating a structurally constrained model of protein evolution, again modelled on the snake puzzle. In this way, we can discuss the analogy between evolutionary concepts and statistical mechanics that facilitates the understanding of both concepts. The proposed examples and literature are accessible, and we provide supplementary material (see 'Data Availability') to reproduce the numerical experiments. We also suggest possible directions to expand the unit. We hope that this work will further stimulate the adoption of games in teaching practice.

# INTRODUCTION

Scientific knowledge is becoming increasingly interdisciplinary, with life sciences being one of the most significant examples. This field has attracted experts from different areas and backgrounds, as foreseen by Schrödinger's seminal book "What is Life?" (*Schrödinger, 1992*). In fact, life sciences offer a very interesting ground for the application of formal methods originated in other disciplines such as physics or informatics, allowing to address

Corresponding author
Alberto Pascual-García,
alberto.pascual.garcia@gmail.com,
apascual@ic.ac.uk

biological questions from different perspectives (*Lazebnik, 2002*). In recent years, the spectacular increase of biological data promoted by high-throughput technologies boosted the development of computational tools for its analysis and classification, an essential task in itself (*Dougherty & Braga-Neto, 2006*). We are witnessing the growth of a theoretical biology with formal parallelism with other disciplines and with the objective to identify the underlying general principles of biological systems.

This scenario challenges the traditional educational programs (*Gallagher et al., 2011*), often reluctant to overcome the boundaries between established disciplines. In contrast, we observe a rapid growth of interdisciplinary publications in the scientific literature, which contributes to bolster and further extend the gap between research and education. As a result, it is common that students with different backgrounds meet in the same postgraduate courses, but they often lack the skills required to work in such an integrative environment. Together with the limited number of teaching hours, this situation constitutes a serious bottleneck to learning. Hence, it is of great importance to find tools that help to bridge the gap between different backgrounds and favour a learning convergence (*Fox & Ouellette, 2013*). Games can help students to assimilate abstract concepts and to address complex problems. There is a growing number of games related with topics as diverse as protein folding (*Cooper et al., 2010*), spin glasses (*Hartmann, 2013*), ecological networks (*Fortuna et al., 2013*), biological data integration (*Schneider & Jimenez, 2012*), geometry (*O'Rourke, 2011*) or scientific induction (*Gardner, 2008*), to name a few.

Here we present an illustrative example where we employed a wooden snake puzzle in a structural bioinformatics course at the Universidad Autónoma de Madrid (Spain). This puzzle can be regarded as a coarse grained (toy) model of a polymer structure and it is strikingly similar to simplified mathematical models proposed in the protein folding literature (see, for instance, *Šali, Shakhnovich & Karplus, 1994*). We propose several exercises accessible to students with a graduate-level background in either biology or physics and notions in programming.

Our goal consists in giving concreteness to the subjects presented in the course through a physical object, and our experience in this sense is very positive. This example allows us to discuss the first steps in the modeling process, i.e., the definition of the system and its epistemological and practical consequences, a discussion that is often neglected. Relying on a physical object allows us to provide a first intuitive contact with the different subjects that we treat, ranging from computational techniques, such as protein structure alignment algorithms, to theoretical concepts, such as protein folding and evolution. We argue that starting from these examples allows the lecturer to introduce more complex problems, in which real examples might be considered. We suggest along the unit different questions that may be followed to expand a course.

One important aspect in which we focused in the development of these problems is the intimate relation between physics and evolution. Adapting the famous quote of Theodosius Dobzhansky, the properties of natural proteins only make sense in the light of evolution and, conversely, the properties of protein evolution only make sense considering the constraints imposed by protein physics (*Liberles et al., 2012*). Furthermore, there is a deep analogy between the statistical mechanics in the space of protein conformations, which

governs protein folding, and the statistical mechanics in the space of protein sequences, which emerges from evolution (*Sella & Hirsh, 2005*).

The paper is organized as follows. Firstly, we present the analogy between the snake puzzle and the contact matrix representation of protein structures, and we propose a simple algorithm to solve the puzzle. The solutions are then adopted as a set of representative model structures, on which we propose computational exercises focused on evolutionary concepts. Throughout the paper we suggest several discussions that are stimulated by the analogy between the computational results and real proteins, proposing chosen references that are easy to handle by postgraduate students. Finally, we provide as Supplementary Material (see 'Data Availability') the input data needed to reproduce all the numerical experiments and the source code to execute some of them.

## PROPOSED EXERCISES

### Puzzle description: visualization and analogies with protein structures

The snake puzzle consists of a wooden chain of $N$ blocks that must be folded into a cube. Each element of the chain, except the first and the last one, is linked to two others. Three linked units are constrained to form either a straight line or a turn that extends into two dimensions. There are at least two versions of the puzzle in different sizes, available in retail stores and online shops. The first version is a 27-unit puzzle that folds into a cube of side 3, and the second one is a 64-unit puzzle that folds into a cube of size 4. In accordance with the nomenclature used in the literature on polymer physics, we will refer to the former as 27-*mer* model and as 64-*mer* to the latter. The exercises proposed in the following sections are based on the 64-*mer* puzzle, which is more complex while remaining computationally tractable. Our proposal is inspired by the similarity between this puzzle and the lattice models of protein structures studied in the literature (the relevant references will be presented all throughout the text).

The number of possible polymer conformations on the cubic lattice is huge. Taking into account the self-avoiding condition that two units cannot occupy the same cell, each new unit can be placed in at most 5 different cells. Each time that the length increases the number of conformations is multiplied by a roughly constant factor, leading to an exponential increase. Numerical computations show that the number of self-avoiding walks on the cubic lattice scales as $N^\gamma \mu^N$ for large $N$, with $\mu \approx 4.68$ (*Bellemans, 1973*), close to the maximum possible value, so that the number of conformations of a 64-*mer* is of the order of $\sim 10^{51}$. The enormousness of these numbers offers the opportunity to introduce the well known Levinthal's paradox (*Levinthal, 1969*) as a starting point for a course focused on protein folding.

In lattice models of protein folding it is often assumed that folded proteins are represented by maximally compact conformations. In the case of the 64-*mer*, these are conformations that occupy the $4 \times 4 \times 4$ cube. This number also increases exponentially, although with a smaller exponent whose asymptotic value can be computed analytically, for instance it is $\mu = 5/e \approx 1.84$ on the cubic lattice (*Pande et al., 1994*). Thus the number is huge

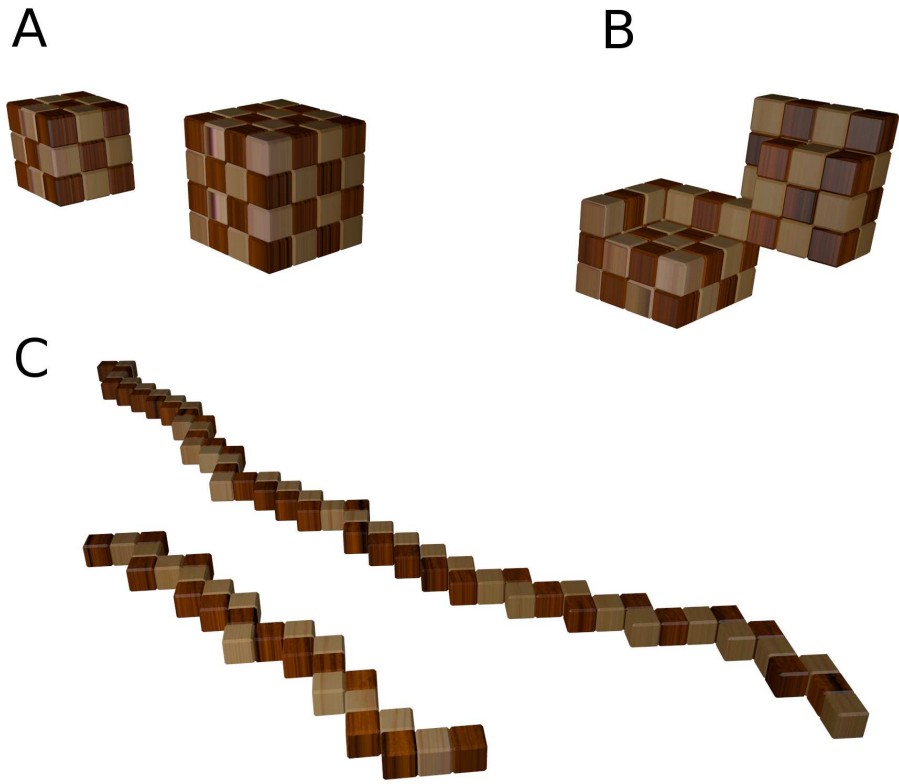

**Figure 1** **Conformations of the 27-*mer* and 64-*mer*.** (A): Maximally compact conformations. (B): Partly open conformation that illustrates the similarity between the model and two protein domains connected through a hinge. (C): Fully extended conformations, where one can see the consecutive rigid fragments of size 2 and 3 (for the 27-*mer*) and size 4 (64-*mer*).

(of the order of $10^{19}$ for the 64-*mer*). An algorithm to generate exhaustively all maximally compact conformations of the 27-*mer* model was proposed in *Shakhnovich & Gutin (1990)*.

With respect to these numbers, the puzzle presents a huge reduction of the conformation space due to the constraints on the direction that each step can take. Indeed, linearly arranged consecutive units can be regarded as a single rigid fragment of size 2, or 3 (27-*mer*) and 2, 3, or 4 (64-*mer*). These rigid blocks can be easily seen in Fig. 1C. The number of fragments is much smaller than the number of units. In addition, two consecutive rigid fragments have only four self-avoiding conformations (two consecutive fragments cannot extend in the same dimension). Consequently, due to these constraints the 64-*mer* puzzle has access to only $\sim 10^{12}$ conformations. Nevertheless, it is still remarkably difficult to solve it without computational help.

This reduction in the order of magnitude of the number of conformations offers an opportunity to discuss how the secondary structure of proteins limits the number of different folded structures, and how secondary structure prediction helps predicting protein structure from sequence. Furthermore, different fragments of the puzzle are sometimes assembled into higher order structures reminiscent of structures found in real proteins. For instance, the 64-*mer* has two consecutive blocks of size 4 that cannot be folded unless they are placed in parallel next to each other. This configuration is remarkably similar to the

naturally occurring *beta sheet* secondary structure. The constraints that this folding entails subdivide the puzzle into two structurally independent modules joined by what resembles a *protein hinge* (see Fig. 1B). On the other hand, there are long regions of consecutive blocks of length 2 that can provide the opportunity to discuss the role of more flexible regions in real proteins such as large loops or intrinsically unstructured regions.

## Solutions of the puzzle

After this preliminary descriptive analysis of the puzzle we propose the first computational exercise, which consists in programming a search algorithm that generates all the maximally compact conformations (i.e., the solutions of the snake puzzle folded in a cube). The solution that we describe in 'Materials and Methods' starts from one end of the chain and exhaustively builds all the conformations of the blocks of the puzzle that fit into a cube of size 3 (for the 27-*mer*) or 4 (64-*mer*). We provide the source code along with the algorithm as Supplementary Material (see 'Data Availability') to the present paper.

It has to be noted that the time taken by our search algorithm grows exponentially with the number of fragments, in accordance with a recent work that shows that the snake puzzle is an NP-complete problem (*Abel et al., 2013*), i.e., loosely speaking, a problem that cannot be solved exactly by any algorithm whose computation time grows only polynomially with system size. Thus, while it would be impossible to apply this algorithm to much larger systems, modern computers are readily capable of handling the calculations for the 64-*mer*. Stochastic algorithms can find solutions in shorter times at the expense of an exhaustive exploration, and they are of interest in the context of protein folding.

To characterize the bottlenecks of the search algorithm, we monitor in which regions of the puzzle the exhaustive search spends more time. In Fig. 2 we plot the histogram of the number of times a fragment is visited during two exhaustive searches that start from the two ends of the chain. To understand the relationship between the search and the constraints imposed by the fragments, we depict in the histogram the position of the rigid fragments of size larger than two. We observe that there is an intense search around the regions where many consecutive small fragments accumulate. On the other hand, the search is drastically reduced when the algorithm finds large consecutive fragments (see, for instance, the two rigid fragments of size 4 separated by a fragment of size 2) because there are few possible conformations compatible with valid spatial coordinates. It is interesting to observe that the number of steps needed to find the solutions varies depending on the end of the chain from which the algorithm starts. This observation suggests that the puzzle may be solved faster if we start the algorithm from regions with more restrictive constraints.

Through the exhaustive search, we find eight solutions for the 64-*mer* puzzle that we show in Fig. 3. These solutions will be labelled hereafter as S1, S2, …, S8. The students can be requested to visually inspect the solutions through standard visualization software such as PyMOL (*DeLano, 2002*) or VMD (*Humphrey, Dalke & Schulten, 1996*). For a review on software for protein structure visualization see *O'Donoghue et al. (2010)*. For this exercise, it is necessary to transform the format of the files with spatial coordinates. This gives us the opportunity to discuss the different file formats with their flaws and

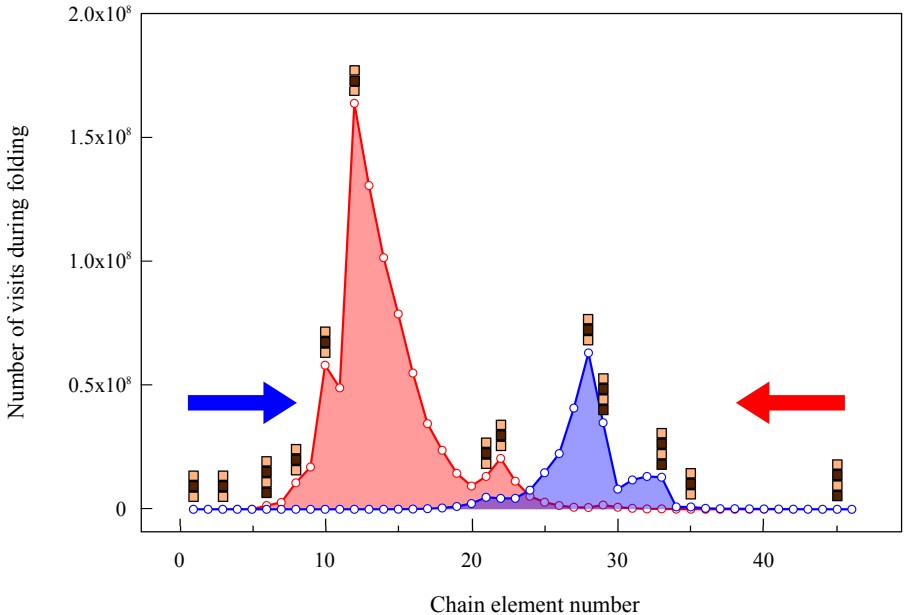

**Figure 2   Number of times that a given rigid fragment is visited by the folding algorithm.** The search starts either from the first (blue histogram) or from the last fragment of the chain (red) and explores all maximally compact conformations. Positions with blocks of length larger than two are depicted over the histograms.

advantages, and to stress the importance of format standardization in bioinformatics (*Gibas & Jambeck, 2001*).

## The contact matrix

At this point, it is convenient to introduce a reduced representation of protein structures that arises naturally in the context of lattice polymers, and that will play an important role in the following: the contact matrix. This binary matrix has value $C_{ij} = 1$ if two units $i$ and $j$ contact each other in space in the polymer conformation, and $C_{ij} = 0$ otherwise:

$$C_{ij} = \begin{cases} 1 & if \ d_{ij} \leq d_0 \\ 0 & otherwise \end{cases} \tag{1}$$

Bonded units ($j = i \pm 1$) are excluded from this count because they contact each other in all conformations. In lattice polymers, the condition for contact is that two units are nearest-neighbours in the lattice, i.e., $d_0$ is the lattice space. We provide the contact matrices of the eight solutions of the snake puzzle in the Supplementary Material (see 'Data Availability') since they are needed to perform the exercises that we describe in the following. Figure 4 shows the four different types of locations that a monomer can occupy within the maximally compact structure and the number of contacts that it has in each case. The same figure also shows an intermediate conformation of two similar solutions, S1 and S2, depicting only their common structure and the first different fragment. The associated contact matrices are represented below, highlighting the differences determined by the alternative positions of the last fragment.

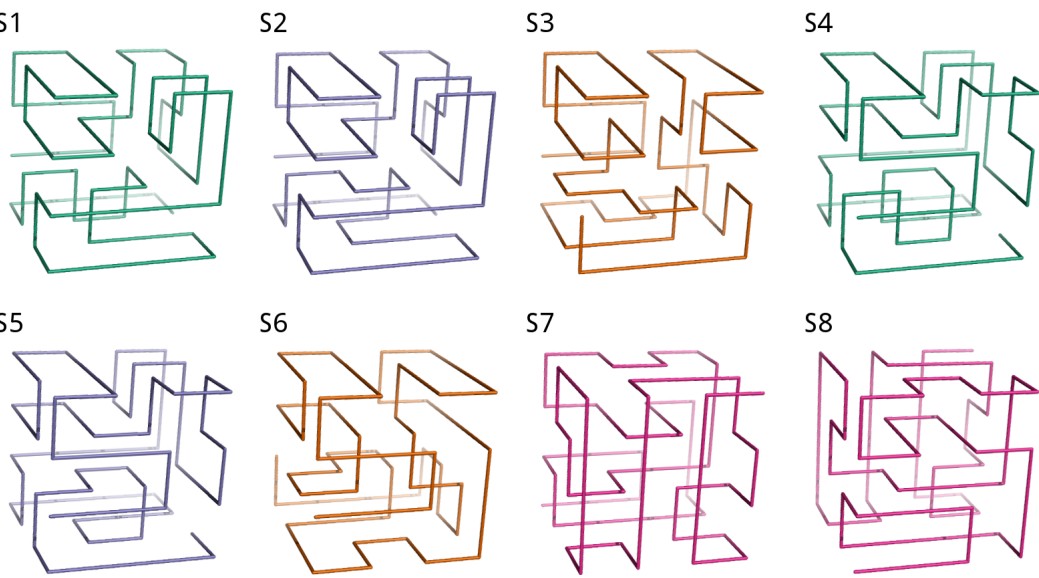

**Figure 3** **The eight solutions of the 64-*mer* puzzle.** Although the solutions cannot be superimposed, they represent four pairs of mirror images, analogous to naturally occurring enantiomers of chemical substances. In the figure, each pair of related solutions are depicted with the same color (S1 & S4, S2 & S5, S3 & S6, S7 & S8). The figure was generated using coordinates files in PDB format, whose documentation can be obtained in http://www.wwpdb.org. The source code provided in the Supplementary Material (see 'Data Availability') outputs PDB files that can be explored using any standard protein structure visualization software.

Contact matrices are also used as a simplified representation of real protein structures. Two residues $i$ and $j$ are considered in contact if their closest not-hydrogen atoms are closer than a threshold, typically $d_0 = 4.5$Å (the main reason to exclude hydrogen atoms is that they are often not reported in PDB files obtained with X-ray crystallography). Pairs are considered in contact only if $|i-j| > 2$, since neighbours along the chain trivially fulfil the contact condition in all conformations. The contact matrix provides a simple visual representation that makes evident secondary structure elements (alpha-helices, appearing as lines of contacts $(i, i+3)$ and $(i, i+4)$ parallel to the diagonal; parallel beta sheets, $(i, i+l)$ and anti parallel beta sheets, $i+l = $ const, appearing as lines perpendicular to the diagonal). Importantly, the contact matrix is independent of the reference frame used to represent the coordinates.

This point gives a good opportunity for a general discussion on the modelling process in biology and its epistemological and practical implications. Protein molecules are extremely complex. They are made of thousands of atoms bound together by quantum interactions. Although not covalently bound, the water solvent is essential in determining the properties of a protein (dynamics, thermodynamic stability, catalytic function...). If we want to make quantitative predictions, we have to reduce this complexity to a simplified model that is amenable to computation. In a statistical mechanical framework, a simplified (mesoscopic) model must be imagined as the result of integrating out some degrees of freedom of the system (the quantum interactions, the water molecules, the hydrogen atoms, the side chains...) and retaining only those that are either most relevant or simplest

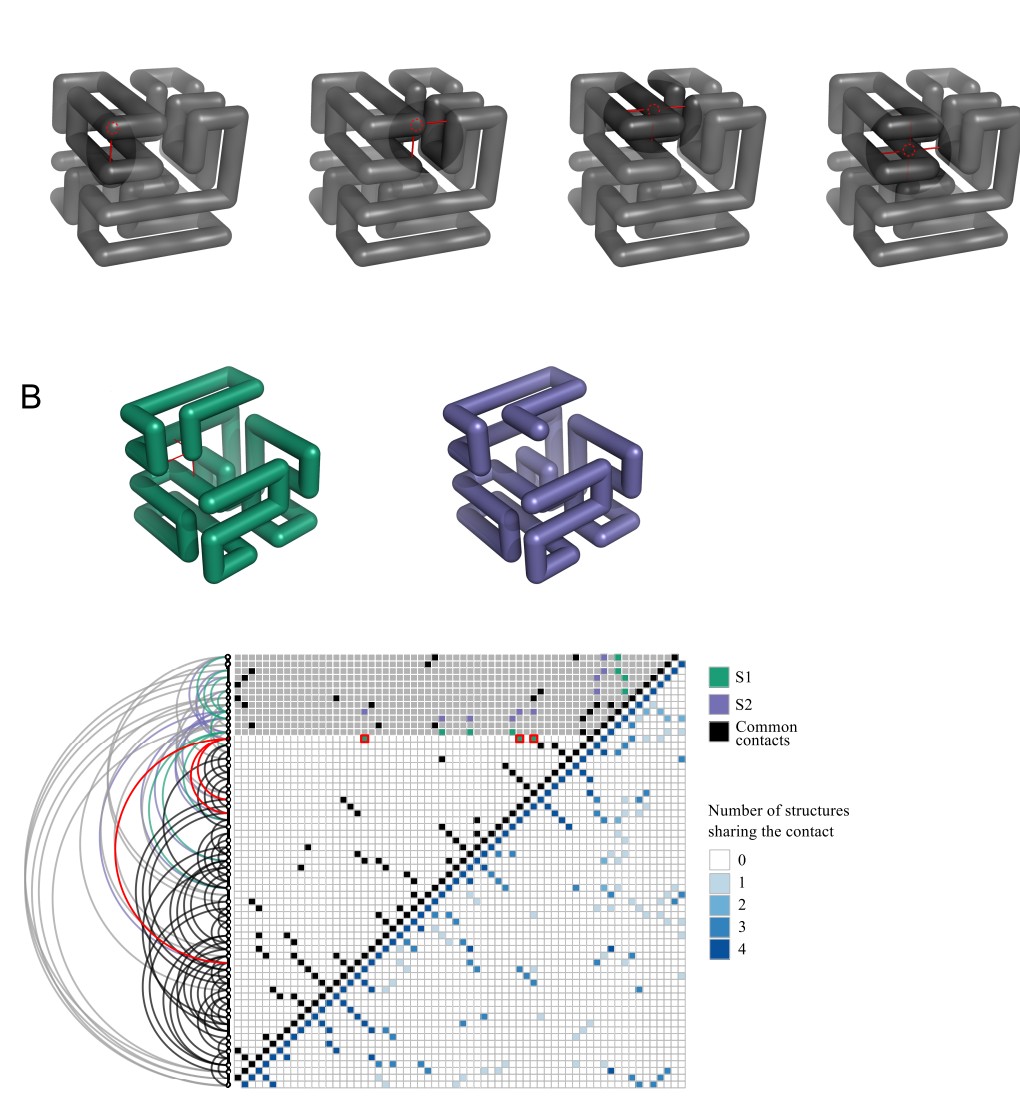

**Figure 4   Contact matrix representation.** (A) Illustration of the four different types of locations where a monomer can be found in a maximally compact structure and the contacts that it will has in each case. A given monomer may have from one to four contacts, with the exception of the first and last monomers, which can present an additional contact up to a maximum of five. (B) Conformations and contact matrices of two similar solutions S1 and S2. In the conformations, for clarity we show only the part that is common to both solutions and the first fragment that is different. The upper triangular matrix shows the contacts of S1 and S2, highlighting in red the contacts present in S1 that are absent in S2 (shown in the conformation S1 in red as well). The grey shaded area in the matrix corresponds to the monomers not shown in the above conformations. The distance in sequence between the contacts can be seen in the arcs representation depicted in the left of the matrix. The lower triangular matrix represents the number of shared contacts among the four non-redundant solutions.

to handle, allowing quantitative predictions. The contact matrix is one such mesoscopic representation. To define it, we need arbitrary choices (the definition of the distance between residues) and parameters (the threshold distance). This is an unavoidable (and indeed desirable) feature of the modelling process.

### Contact energy function

Nevertheless, although the modelling choices may seem plausible, it is important that they are tested a posteriori for their predicting ability, and that parameters are optimized by comparison with experimental data, if it is possible. In the case of protein contact matrices, predictions are obtained from statistical mechanical models that present a simple contact energy function:

$$E(C,A) = \sum_{i<j} C_{ij} U_{ij} \tag{2}$$

where $C_{ij}$ is the contact matrix and $U_{ij}$ is the contact interaction energy between residues $i$ and $j$ that are in contact, which may be imagined as the result of averaging out all other degrees of freedom in a thermodynamic ensemble subject to the constraint that $i$ and $j$ are in contact. Such implicit computation is of course impossible to realize, and researchers adopt two main types of contact free energy functions.

The first type belongs to the broad category of knowledge-based energy functions. In this case, $U_{ij}$ depends on the type of amino acids at positions $i$ and $j$, $U_{ij} = U(A_i, A_j)$, where the sequence $A_i$ denote the amino acid type at position $i$ and $U(a,b)$ is a $20 \times 20$ symmetric interaction matrix derived from large databases of protein structures and sequences, with the aim to rationalize or predict the folding stability of proteins, such as for instance the Miazawa and Jernigan potential (*Miyazawa & Jernigan, 1996*). The second type belongs to the category of structure-based energy functions, which are determined from each experimentally determined protein native structure in such that the native structure has minimum free energy and that the native state is minimally frustrated, i.e., all native interactions (and only them) are stabilizing, and all pairs of atoms that interact in the native state minimize their interaction energy: $U_{ij} = -C_{ij}^{\mathrm{nat}}$, where $C^{\mathrm{nat}}$ is the native contact matrix.

Some structure-based models are constructed as an explicit function of the inter-residue distance $d_{ij}$, $E = \sum_{ij} C_{ij}^{\mathrm{nat}} f_{ij}(d_{ij})$. The minimum frustration principle (*Bryngelson & Wolynes, 1987*) requires that the minimum of each pairwise interaction $f_{ij}$ corresponds to the native distance $d_{ij}^{\mathrm{nat}}$. Structure-based models are used to predict protein dynamics close to the native structure through normal mode analysis (the so-called Elastic Network Model) (*Tirion, 1996*; *Bahar & Rader, 2005*) or to predict the thermodynamics and kinetics of the folding process (*Taketomi, Ueda & Gō, 1975*) and, despite their simplicity, they provide accurate predictions that are favourably compared to experimental observations, as it can be seen in reviews on these subjects (*Bastolla, 2014*; *Chan et al., 2011*).

### Contact order

An important quantity that can be computed from the contact matrix is the Absolute Contact Order (ACO), defined as:

$$\mathrm{ACO} = \frac{\sum_{i<j} C_{ij}|i-j|}{\sum_{i<j} C_{ij}} \tag{3}$$

i.e., the average distance in sequence between pairs of residues spatially in contact. It has been observed that the ACO is negatively correlated with the folding rate of the protein, in such a way that proteins with larger ACO tend to fold slower (*Ivankov et al., 2003*).

Figure 5 represents the contact order for each of the four solutions of the puzzle. In addition, a more compact representation shows the number of contacts per residue. We find the following ACO values: 18, 17, 15 and 14 for S1, S2, S3, and S7, respectively. The analogy with real proteins suggests that structures S1 and S2 fold more slowly, as they present larger contact order. Conversely, the folding dynamics of S7 is expected to be faster and that of S3 would be intermediate. We also note the presence of common contacts in all four solutions, which are related with more constrained regions, e.g., the contacts involving the two consecutive blocks of length 4 mentioned above.

## Structural comparisons
### Contact overlap

Another important application of contact matrices consists in comparing two protein structures. For protein structure comparison, as for sequence comparison, the first step consists in determining an *alignment*, i.e., a correspondence $i_y = a(i_x)$ between the position $i_x$ in protein x and the position $i_y$ in protein y. If the two structures correspond to different conformations of the same protein, the alignment is trivial: $a(i) = i$. Here we assume that this is the case, and we will discuss alignment algorithms below. We can measure the similarity between any two polymer structures $S_x$ and $S_y$ through the *Contact Overlap*:

$$Contact\ Overlap = \frac{\sum_{ij} C_{ij}^{(x)} C_{a(i)a(j)}^{(y)}}{\sqrt{\sum_{ij} C_{ij}^{(x)} \sum_{ij} C_{ij}^{(y)}}}. \tag{4}$$

This measure is zero if the structures do not share any contact, and one if they share all contacts.

We can ask the students to compute the Contact Overlap between all pairs of solutions of the 64-*mer*, which are reported in Table 1. It can be noted that several off-diagonal pairs have overlap equal to one, meaning that their contact matrix is exactly the same. This result is puzzling, since our algorithm guarantees that all solutions are different, in the sense that they cannot be superimposed through rigid body rotations or translations. We can ask the students to explain this fact. The answer is that structures with overlap equal to one are related through a mirror reflection—they correspond to chemical enantiomers. It can be shown (see 'Material and Methods') that mirror images can be excluded by the search algorithm through a suitable control of the axes. In the following, we will only consider one of each pair of enantiomers (S1, S2, S3, and S7) to avoid redundancy. Figure 4 compares the contact matrices of the structures S1 and S7. A summary of the contacts found for all four non-redundant solutions is also shown.

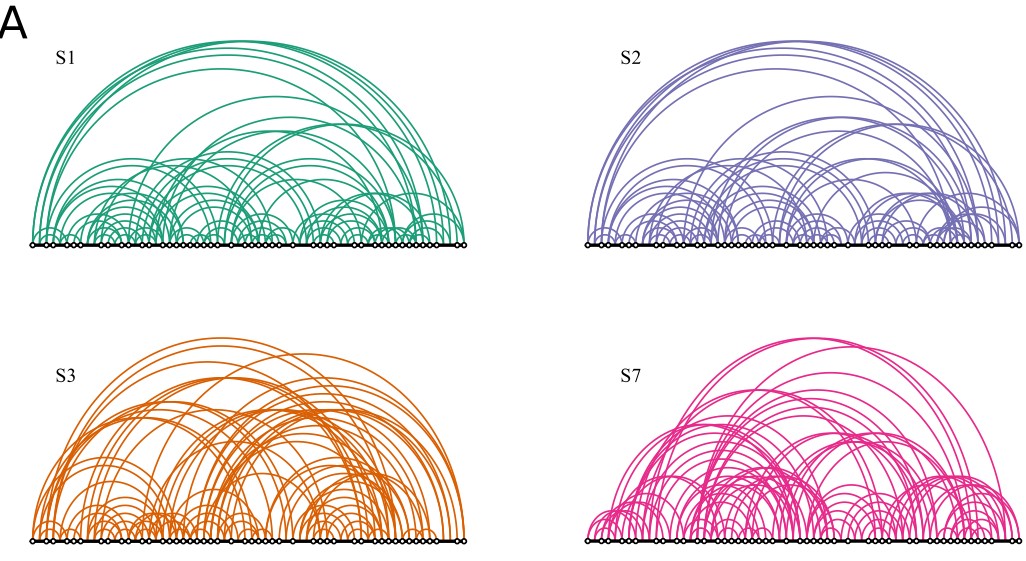

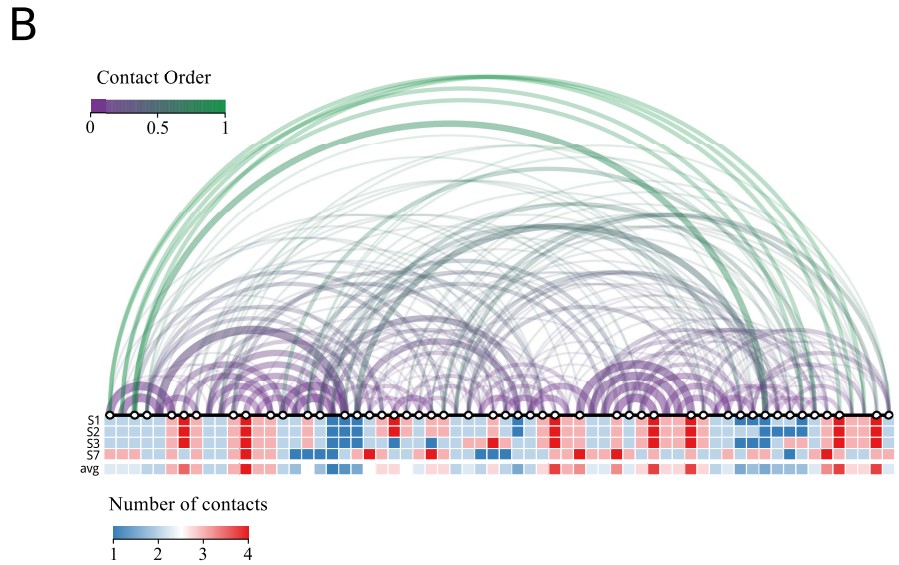

**Figure 5  Network representation of the solutions.** The black line represents the chain and each white dot represents an opposite end of the rigid fragment. The distance between dots is proportional to the length of the fragment. Each position in the chain is connected with other positions if they are in contact in the folded solution. Longer arcs have a larger contribution to the contact order. (A) Representation for the four non redundant solutions. Solutions S1 and S2 have contacts with larger contact orders whereas solution S7 has the shortest. (B) The same representation integrating all four solutions. The width of the arc is proportional to the number of structures sharing the contact. The colour code quantifies the distance between each pair of residues (terms in the numerator of Eq. (3)) normalized by the length of the chain (which is the same for all solutions). We note a region of four contacts on the right half of the chain common to all solutions which corresponds to the peak shown in Fig. 2. The number of contacts per residue is shown for each solution, together with the average. Note that solution S7 has the most dissimilar profile.

**Table 1  Pairwise comparison of the solutions of the snake puzzle through the Contact Overlap.**
Note that there are several off diagonal values with overlap one (perfect similarity) representing non-superposable mirror images.

| Contact Overlap | S1 | S2 | S3 | S4 | S5 | S6 | S7 | S8 |
|---|---|---|---|---|---|---|---|---|
| S1 | 1.00 | 0.88 | 0.41 | **1.00** | 0.88 | 0.41 | 0.16 | 0.16 |
| S2 | | 1.00 | 0.37 | 0.88 | **1.00** | 0.37 | 0.19 | 0.19 |
| S3 | | | 1.00 | 0.41 | 0.37 | **1.00** | 0.15 | 0.15 |
| S4 | | | | 1.00 | 0.88 | 0.41 | 0.16 | 0.16 |
| S5 | | | | | 1.00 | 0.37 | 0.19 | 0.19 |
| S6 | | | | | | 1.00 | 0.15 | 0.15 |
| S7 | | | | | | | 1.00 | **1.00** |
| S8 | | | | | | | | 1.00 |

### Optimal superposition. Root Mean Square Deviation

At this point, it is interesting to compare the properties of the Contact Overlap with another measure of structural distance, the Root Mean Square Deviation

$$\text{RMSD} = \min_R \sqrt{\frac{1}{n} \sum_i \left| \vec{r}_i^{(x)} - R\vec{r}_{a(i)}^{(y)} \right|^2}, \tag{5}$$

where $\vec{r}_i^{(x)}$ indicates the coordinates of atom $i$ in structure $x$, $|\cdot|^2$ is the Euclidean distance, $R$ denotes a rotation matrix that has to be optimized to find the optimal superimposition, and both $\vec{r}_i^{(x)}$ and $\vec{r}_i^{(y)}$ are translated in such a way that their centers of mass stay at the origin. The above formula for the RMSD helps to avoid the confusion between alignment $a(i)$ and superposition $R$, which is frequent among students.

Instead of comparing interatomic distances between pairs of atoms as the Contact Overlap does, the RMSD compares the distances of individual aligned atoms after optimal superimposition, which is apparently simpler. However, this simplification is obtained at the price to determine the optimal rotation matrix $R$ that minimizes the RMSD. This minimization can be performed analytically through the classical algorithm by Kabsch based on Singular Value Decomposition. Nevertheless, the optimal superimposition is strongly influenced by the aligned atom $i$ that is farther away in the two structures. This determines a trade-off between alignment length and RMSD, since we can decide not to align residues that are too distant, obtaining shorter alignments with smaller RMSD. In this sense, the measure of the RMSD is not uniquely determined unless the alignment is trivial. On the contrary, the Contact Overlap is independent of the superimposition and, apart for the computation time, it is possible in principle to determine the alignment that maximizes the Contact Overlap without any arbitrary choice. It is important to note that the RMSD is expected to be only loosely correlated with distances in energy, since flexible regions can produce large RMSD at a small energetic cost. The same transformation is expected to maintain a large Contact Overlap, since flexible regions are characterized by few contacts (*Wallin, Farwer & Bastolla, 2003*).

### Structure alignments

After discussing the notion of spatial superposition, we can now discuss the concept of structure alignment. This is easier adopting the Contact Overlap. In this context, the optimal pairwise alignment $a(i)$ can be defined as the alignment that maximizes the Contact Overlap Eq. (4). Since protein structure is conserved through evolution, we expect that pairs of aligned residues associated through the correspondence $a(i)$ are likely to be evolutionarily related (homologous).

It is clear that the algorithm for finding the optimal structure alignment given a scoring scheme will be more complex than the algorithm for finding the optimal sequence alignment that depends only on the similarity between the individual amino acids at positions $i$ and $a(i)$. It can be shown that the structure alignment based on the Contact Overlap is an NP-complete problem, which, loosely speaking, means that no algorithms that runs in polynomial time can guarantee to find the optimal solution in all cases, while for pairwise sequence alignments the optimal solution can be found in polynomial time through dynamic programming. Nevertheless, good heuristic algorithms exist, such as the Monte Carlo algorithm implemented in the structural alignment algorithm Dali (*Holm & Sander, 1993*), which is conceptually related to the optimization of the Contact Overlap.

On the other hand, there is no structure alignment method based on the minimization of the RMSD. In fact, when we superimpose two aligned proteins, the optimal superimposition is strongly affected by pairs of residues $i, a(i)$ with large distance, typically located in flexible regions. Consequently, the optimal superimposition $R$ may locate far apart atoms in the structural cores of the two proteins for the sake of improving the alignment of outliers. Therefore, for distantly related proteins, it is preferable to superimpose only the structural cores, constituted by pairs of residues whose interatomic distance is smaller than a threshold. This introduces a trade-off between the alignment $a$, the structural superimposition $R$ and the core definition, that must be sorted out through the choice of the threshold and some kind of heuristic, such as in the structure alignment algorithm Mammoth (*Ortiz, Strauss & Olmea, 2002*).

Instead of applying a fixed threshold to define the core, the program TMalign computes the average distance between unrelated residues as a function of the protein length, and adopts this function in such a way that only pairs that are closer than expected by chance give a positive contribution to the structural similarity (*Zhang & Skolnick, 2005*). TM-align is one of the most reliable structure alignment programs based on rigid superimposition. Notably, methods have been proposed that consider flexible superposition in which different rigid body transformations are applied to different protein domains (e.g., ProtDeform *Rocha et al., 2009*).

### Protein structure classification

Protein structure comparison may also be a convenient point for introducing protein structure evolution. We will base our analysis on protein superfamilies, groups of *bona fide* evolutionarily related protein domains that are structurally similar and can be found in structural classification databases such as SCOP (*Murzin et al., 1995*) and CATH (*Orengo et al., 1997*).

Comparing proteins within a superfamily, it was found that protein structure and protein sequence divergence are linearly related. This result has been established based on the RMSD of aligned and superimposed protein cores (*Chothia & Lesk, 1986*) and later extended to other measures of structural change, allowing to quantify the statement that protein structure is more conserved than protein sequence (*Illergård, Ardell & Elofsson, 2009*).

In this context, our group introduced the Contact Divergence (CD) (*Pascual-García et al., 2010*), a measure of structural divergence derived from the Contact Overlap, and analogous to the Poisson distance between protein sequences, in such a way that the CD is expected to be related with the time during which proteins diverged in evolution. By comparing the Contact Divergence in protein structure space with the Poissonian distance between the corresponding sequences, we found that structure evolution is slower than sequence evolution by a factor that ranges from 0.24 to 0.37 for different superfamilies (*Pascual-García et al., 2010*). Importantly, proteins that conserve exactly the same molecular function appear to be limited in their CD, which suggests that homology modelling can be rather successful in the case of function conservation, while structure evolution is more irregular and the CD explodes for proteins that change their molecular function.

The notion that protein structures may be largely different within the same superfamily leads to challenge the concept that protein structure space is organized in discrete regions called "folds" that represent well-defined equivalence classes of protein structures, an idea that underlies most structural classifications of proteins (*Orengo et al., 1997*; *Murzin et al., 1995*; *Holm & Sander, 1997*). More recently, it is gaining force an alternative view that sees protein structure space as constituted by discrete folds only at very high similarity, while at low similarity structure space is continuous and should be described as a network rather than as a tree. (*Dokholyan, Shakhnovich & Shakhnovich, 2002*; *Pascual-García et al., 2009*; *Skolnick et al., 2009*; *Sadreyev, Kim & Grishin, 2009*).

## Sequence-structure relationship and protein designability

The key to protein evolution is the relationship between the protein sequence, which evolves through time, and protein structure and function. The very simple models presented in this unit can constitute a suitable introduction for such a subject.

We can define the sequence structure relationship within the contact energy model of Eq. (2) where a contact matrix $C_{ij}$ represents the mesoscopic state made of all structures with contact matrix $C_{ij}$, and Eq. (2) represents its effective free energy, with all other degrees of freedom averaged out. In principle the effective energy depends on the temperature and the state of the solvent, but these dependencies will be kept implicit. We assume that the contact interaction energies depend on the protein sequence as $U_{ij} = U(A_i, A_j)$, where $A_i$ denotes the amino acid type at position $i$. To keep things simplest, we consider the HP model that groups the amino acid in two types, hydrophobic (H) and polar (P). More realistic models consider the twenty natural amino acid types and contact interaction energies with 210 parameters. This simple HP model is sufficient to reproduce many qualitative features of protein folding. Indeed, hydrophobicity is considered as the main force that

stabilizes folded proteins. We adopt the parametrization suggested by *Li et al. (1996)*, with $U(H,H) = -2.3$, $U(H,P) = -1$, and $U(P,P) = 0$, which satisfy the following physical constraints: (1) more compact conformations with more contacts have lower energies, (2) hydrophobic monomers are buried as much as possible and (3) different types of monomers tend to segregate, which is achieved if $2U(H,P) > U(H,H) + U(P,P)$.

For a given sequence, the ground state of this model is the contact matrix with the lowest effective energy among all physically feasible contact matrices that satisfy the strong conditions of chain connectivity, excluded volume (the self-avoiding interactions) and secondary structure (if we consider real protein structures). While the number of all possible contact matrices is huge (of the order of $2^{L^2}$), most of them are unfeasible since the number of self-avoiding walks increases only exponentially with the number of monomers $L$. As a model of feasible contact matrices, we can consider the contact matrices of compact self avoiding walks on the cubic lattice or the contact matrices of real protein structures. Here, for the sake of simplicity, we will limit our computations to the solutions of the snake puzzle. This is analogous to the toy model of protein folding based on the maximally compact structures on the cubic lattice introduced by Shakhnovich and coworkers in the 90's (*Shakhnovich & Gutin, 1990*).

As suggested by several works, we identify the native state of a protein sequence with the ground state of its effective energy (more realistic models also impose conditions on the stability with respect to alternative compact conformations, as we will see in the following). This allows us to define the designability of a given structure as the number of sequences for which this structure is the ground state.

More designable structures are expected to be more resistant to sequence mutations. Mutational robustness has been proposed to be important to reduce the deleterious effects of erroneous protein translation in the ribosomes (*Drummond & Wilke, 2009*) and to mitigate the mutation load determined by unviable offsprings generated under strong mutation rate (*Bloom et al., 2007*; *Lauring, Frydman & Andino, 2013*). Furthermore, it has been shown that robustness favours the evolution of new protein functions (*Soskine & Tawfik, 2010*). Similar considerations motivated the computational study by *Li et al. (1996)*, who considered as feasible states the contact matrices of the maximally compact self avoiding walks of 27 steps on the cubic lattice, whose number is $\sim 10^5$ (*Shakhnovich & Gutin, 1990*) and evaluated their designability in the space of the $2^{27}$ sequences of 27 H and P.

Here we propose that the students reproduce this computation, adopting as feasible structures three of the solutions of the snake puzzle: S1, S3 or S7. We discard the solution S2 because it is very similar to S1 (see the Contact Overlap values in Table 1) and it should be considered part of its native valley rather than an alternative conformation.

Figure 6 shows the results for the three structures considered for an increasing number of sequences (see 'Materials and Methods'). *Li et al. (1996)* observed that the distribution for the number of structures with $S$ sequences decays exponentially with $S$, which means that there are few structures with high designability and many with low designability. Of course three structures are too few to test this behaviour, but it is interesting that there are

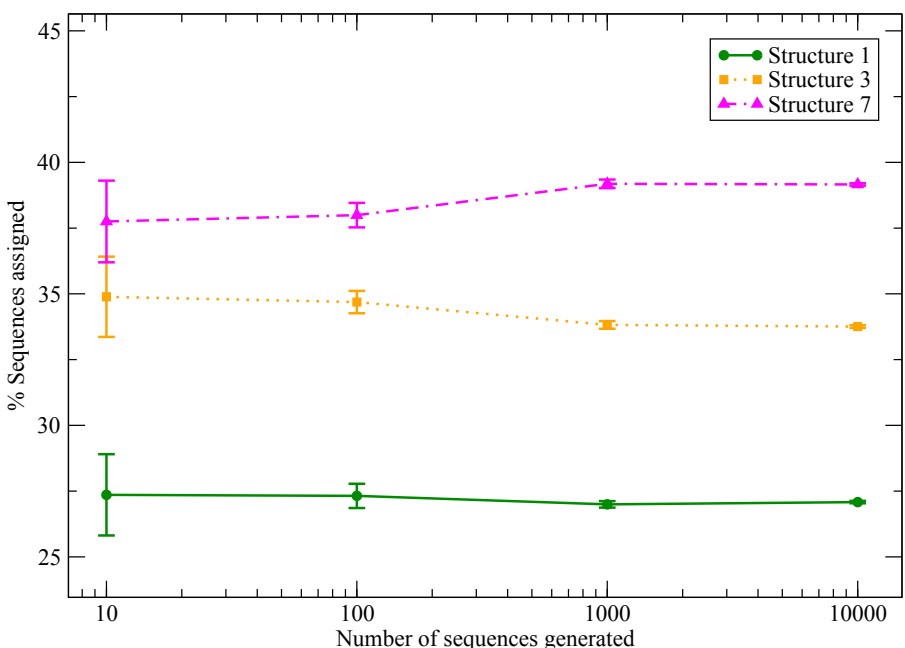

**Figure 6  Designability of selected structures for an increasing number of random sequences.** We observe significant differences between the structures that tends to a well defined limit when the number of sequences increases.

important differences between the three structures, and S7 stands out as more designable than the rest.

This exercise allows us to emphasize the importance to assess the statistical significance and statistical errors in computational studies. The differences that we observe are relevant only if they are statistically significant, i.e., if the probability that they arise by chance is lower than a predetermined threshold (typically, 0.05). We propose two methods to verify that they are indeed significant. The first one, more rigorous, is based on the binomial distribution: given any structure $a$, we can compute the probability that the number of sequences assigned to $a$, $S_a$ over a total of $S$ that were tested, is the result of $S$ tests with success probability $1/3$, as if all structures have the same probability. If all these probabilities are small, then the designabilities are significantly different. The second method, easier to apply in general, consists in computing the standard error of the mean of the designability $p_a$, $s_a = \sqrt{p_a(1-p_a)/S}$, and performing an unpaired $Z$-test with $Z = (p_a - p_b)/\sqrt{s_a^2 + s_b^2}$ for all pairs of structures. With 0.05 as significance threshold, the binomial test finds the difference from equal frequencies not significant for $S \leq 100$ (except for S1) but significant for $S \geq 1,000$ for all structures. The $Z$ test yields the same result (in this case, S1 is not significant for $S = 100$).

## Misfolding stability and energy gap

It has been noted in several works that the analogy between the polymer model and protein folding only makes sense if the ground state structure, identified as the native state, is

much more stable than alternative structures, which collectively represent the misfolded ensemble.

One of the first parameters used to assess the stability of the putative native state has been the energy gap between the putative native state and the misfolded conformation with minimum energy among those not in the attraction basin of the native state (*Sali et al., 1994*). If this gap is small, the native state lacks thermodynamic stability, since a small change in the parameters of the effective energy function, corresponding to a change of temperature or pH, would produce a different ground state; it lacks stability against mutations in the sequence, for similar reasons; and it is very difficult to reach kinetically, since the dynamics of the model polymer can become trapped in low energy conformations outside the native basin. In their work cited above, *Li et al. (1996)* found that more designable structures have a larger energy gap, and that the energy gap averaged over the sequences assigned to a given structure clearly separates the maximally designable structures from the rest. In our toy model there is one "native" and only two "misfolded" conformations. We denote by $E(C^i, A^k)$ the effective energy of sequence $A^k$ in structure $i$, and we define the energy gap as the effective energy difference between the "native" structure and the alternative conformation with lowest energy

$$\delta E(C^i, A^k) = E(C^i, A^k) - \min_{j \neq i} E(C^j, A^k). \tag{6}$$

This quantity has then to be averaged over all sequences $A^k$ assigned to structure $C^i$, which we denote with an overline:

$$\overline{\delta E(C^i)} = \overline{E(C^i, A^k) - \min_j E(C^j, A^k)}(j \neq i). \tag{7}$$

The values of the average energy gap are shown in Fig. 7. We can see that, consistently with the results reported by *Li et al. (1996)* the structure S7 has both the highest designability and the largest energy gap, and that these differences are significant. We can then ask students whether this correlation between designability and energy gap is surprising or it should be expected based on the definitions.

Another measure of the stability of the misfolded ensemble, more standard under the point of view of statistical mechanics, is its free energy, which can be evaluated through some simplified statistical mechanical model.

We assume that the protein can exist in three macroscopic states: the native state (the attraction basin of the contact matrix with minimum effective energy, which can be described for instance through the structure-based elastic network model mentioned above), the unfolded state (analogous to the self-avoiding-walk model, with large conformation entropy and effective contact energy close to zero), and the misfolded state (alternative compact conformations dissimilar enough from the native structure). We describe each of these states (native, misfolded and unfolded) by a free energy which, at constant pressure and volume, is given by the difference between the total energy $E$ and the conformational entropy $S$ multiplied by the absolute temperature, i.e., the Helmholtz free energy $G = E - TS$. Furthermore, the Helmholtz free energy is proportional to the logarithm

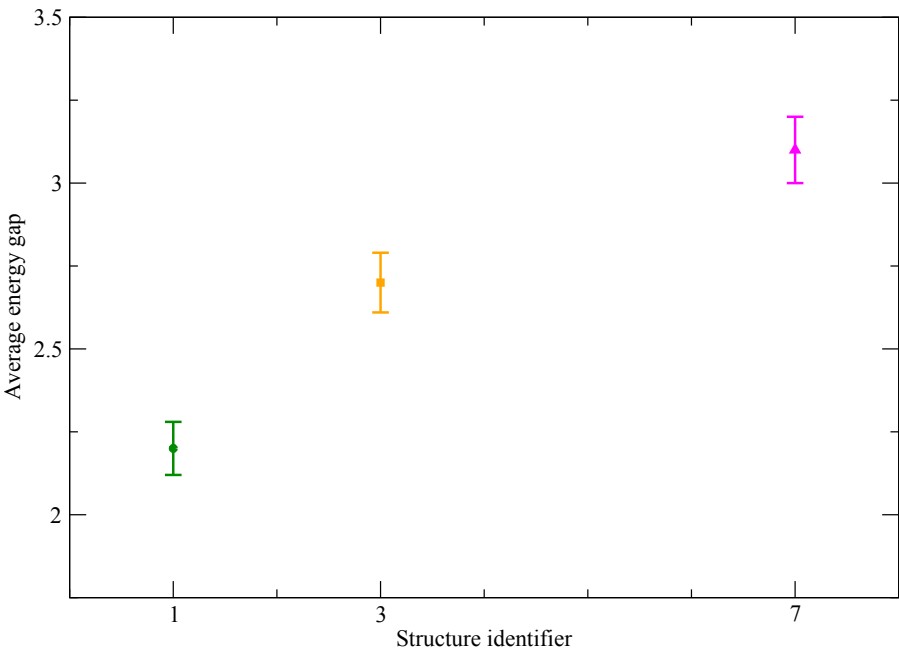

**Figure 7** **Average energy gap for the three chosen structures, S1 (green), S3 (orange) and S7 (pink).** The structure S7 shows a significantly higher value than the others.

of the partition function $G = -k_B T \log Z$ of the canonical ensemble, an important equation that permits to relate macroscopic thermodynamic quantities with the microscopic details encoded in the partition function $Z$. In the following, we show how the free energies of the different states can be estimated, and how we can define the folding free energy of the model polymer $\Delta G$ as the difference between the free energy of the native state and that of the unfolded plus misfolded state (the non-native state). This gives us the opportunity to stress that both unfolding and misfolding are important and should be taken into account.

We approximate **the free energy of the native state** as its effective contact energy, $G_{nat}(A) = \sum_{i<j} C_{ij}^{nat} U(A_i, A_j)$ where $C_{ij}^{nat}$ indicates the contact matrix of the native structure. Note that we should also take into account the conformational entropy of the native state, derived from the volume in phase space of the structures that share the native contact matrix, but this conformational entropy is expected to be similar to that of misfolded contact matrices (*Karplus, Ichiye & Pettitt, 1987*), so that it contributes little to $\Delta G$.

**The free energy of the unfolded state** is approximated as its conformational entropy, which is proportional to the number of residues (for instance, for a self avoiding walk of $L$ steps with $\mu^L$ conformations the free energy would be $G_{unf} = -k_B T L \log \mu$). For a more realistic model, we should consider the number of degrees of freedom that are not frozen (typically, the torsion angles phi and psi of the main chain and chi of the side chains) and the limitations to their movement imposed by atomic interactions, in particular the repulsion that motivates the self-avoiding-walk model. An approximation that is often used is $G_{unf}(A) = T \sum_i s(A_i)$, where the conformational entropy of residue $A_i$, $s(A_i)$, is an empirical value that mainly depends on its number of chi angles.

The term that is most difficult to estimate, and that is often neglected in computational models despite its importance, is **the free energy of the misfolded state**. In the context of the contact energy model, this term can be estimated exploiting the analogy with the random energy model (REM) (*Derrida, 1981*). For an analysis of the selective pressures dictated by stability against misfolding and predicted through the REM approximation, see *Minning, Porto & Bastolla (2013)*. The first term of the free energy expansion of the REM is just the average effective energy of alternative conformations, which is very simple to compute. We denote by $\langle \cdot \rangle$ the average over the ensemble of alternative compact conformations, and we approximate the free energy of the misfolded ensemble as the average energy of misfolded contact matrices,

$$G_{\mathrm{misf}}(A) = \sum_{i<j} \langle C_{ij} \rangle U(A_i, A_j) \approx \frac{2N_C}{L(L-1)} \sum_{i<j} U(A_i, A_j) \qquad (8)$$

where $\langle C_{ij} \rangle$ denotes the average value of the contact between residues $i$ and $j$ in the misfolded ensemble, and we adopt the approximation that all such average contacts are equal and the average number of contacts of misfolded structures is equal to the number of contacts of the native structure, $\sum_{i<j} \langle C_{ij} \rangle = N_C$. In real polymers, $\langle C_{ij} \rangle$ decreases with the sequence distance between the two residues in contact, $|i-j|$, as predicted by polymer physics, and as it can be verified through a statistical analysis of the PDB, but we neglect this dependence for simplicity.

Next, we combine the misfolded and unfolded states into a single non-native state containing both states separated by an energy barrier, which we describe with the partition function $Z_{\mathrm{nonat}} = \left(e^{-\beta G_{\mathrm{misf}}} + e^{-\beta G_{\mathrm{unf}}}\right)$. We define the folding free energy $\Delta G$ as the difference between the native and non-native free energy states,

$$\Delta G(A) = G_{\mathrm{nat}} + k_{\mathrm{B}} T \log(Z_{\mathrm{nonat}}) \qquad (9)$$

where $\beta = 1/k_{\mathrm{B}} T$ is the Boltzmann factor. Finally, we assume that the free energy of the misfolded ensemble is much more negative than that of the unfolded ensemble, which will be neglected. Thus, we estimate the folding free energy as

$$\Delta G(A) = \sum_{i<j} C_{ij}^{\mathrm{nat}} U(A_i, A_j) - \frac{2N_C}{L(L-1)} \sum_{i<j} U(A_i, A_j) \, . \qquad (10)$$

We will adopt the above free energy for the model of protein evolution described in the next section.

In the context of evolution, we call *positive design* the selective forces that make the first term of the above equation more negative, increasing the stability of the native state against that of the unfolded state, and *negative design* (*Berezovsky, Zeldovich & Shakhnovich, 2007*; *Noivirt-Brik, Horovitz & Unger, 2009*) the selective forces that make the second term of the above equation more positive, decreasing the stability of the misfolded ensemble (which, as it is important to note, is independent of the native state). Protein evolution pursues both positive and negative design at the same time. However, positive and negative design may have contrasting requirements. Hydrophobicity is recognized as the main

force underlying protein folding. Increasing hydrophobicity favours positive design, since the native energy becomes more negative, but it contrasts negative design, since the free energy of the misfolded ensemble becomes negative as well (and at a faster pace than the native energy, since the REM free energy has a term proportional to the mean square energy of alternative conformations). Evolution has to finely balance these two selective forces, and the balance depends on the mutation bias, i.e., on whether mutation favour hydrophobic or polar amino acids (*Méndez et al., 2010*). We will illustrate these issues and their interplay with other evolutionary forces such as mutation bias and population size with the computational exercise proposed in next section.

## Structurally constrained sequence evolution

The main point we address here is that the thermodynamic properties of folding observed in natural proteins are a consequence of the evolutionary process, consisting of mutation and selection. With the aim to "bring molecules back into molecular evolution" (*Wilke, 2012*), we will show how protein evolution is jointly constrained by evolutionary parameters (mutation bias, population size, temperature) and the requirement to fold into the target native structure.

From a didactic point of view, protein evolution offers a valuable opportunity to illustrate the essence of statistical mechanics in a way that is intuitive for biologists. The evolutionary model described here can be interpreted as a statistical mechanical model in the space of the protein sequences constrained by the requirement of the stability of the native state.

Our evolutionary model considers a genetically homogeneous population, i.e., we assume that the mutation rate is very small. It is important to distinguish between a mutation, which is a microscopic event that affects individuals, and a mutation that becomes fixed in the population (often called substitution, although this term would only be used for amino acid or nucleotide changes and not for insertions and deletions), which is the macroscopic event that interests us here. Every time a mutation occurs in a sequence $A$, it may either disappear or get fixed in the population with a probability $P_{\text{fix}}$ that depends on its fitness relative to the wild type and on the population size $N$,

$$P_{\text{fix}}(A \to A') = \frac{\left(f(A)/f(A')\right) - 1}{\left(f(A)/f(A')\right)^N - 1} . \tag{11}$$

Where $A'$ is the mutant sequence. It is often assumed that the fitness depends on the stability of the native state and is proportional to the fraction of proteins that are folded at the thermodynamic equilibrium (*Goldstein, 2011*; *Serohijos & Shakhnovich, 2014*)

$$f(A) = \frac{e^{-\beta \Delta G(A)}}{1 + e^{-\beta \Delta G(A)}} \tag{12}$$

where $\Delta G(A)$ is modelled for instance as in Eq. (10) and $\beta = 1/k_{\text{B}}T$ is the inverse of temperature. Although other properties—in particular the dynamics of the protein, its capacity to interact with other proteins, its catalytic rate and so on—are arguably more important than its stability, they are more difficult to model, and the stability is a necessary

prerequisite at least for the large number of proteins that must be folded in order to function (i.e., excluding natively unfolded proteins).

If the mutation is disadvantageous, i.e., $f(A') < f(A)$, corresponding to a lower stability, the probability that it becomes fixed is exponentially small with population size, but it is non-zero if $\log f(A) - \log f(A')$ is of the order of $1/N$. In other words, the smaller is the population, the more tolerant it is to mutations that decrease the fitness. Analogously, a thermodynamic system is more tolerant to changes that increase its energy the higher is its temperature.

This analogy can be made more meaningful if we consider the evolutionary trajectory as a random walk in the space of the possible sequences (*Sella & Hirsh, 2005*). More precisely, it is a Markov process, since the probability to jump to sequence $A'$ at time $t+1$ only depends on the sequence visited by the population at the previous time $t$. In our opinion, the image of a homogeneous population jumping in the space of possible genotypes through random mutations subject to an acceptance probability—that depends on fitness differences and on population size—makes more intuitive the abstract concept of a Markov process.

A remarkable property of Markov processes is that, under not very restrictive mathematical conditions, after a large enough time they converge to a limit distribution over their phase space. This limit distribution is easy to compute if the Markov process has a mathematical property called detailed balance, or reversibility in the molecular evolution literature and is the basis of the utility of Markov processes in statistical mechanics. In fact, the Boltzmann distribution in conformation space, which can be expressed mathematically as $P(C) = \frac{1}{Z}\exp(-\beta E(C))$, being $E(C)$ the energy of the conformation $C$ and $Z$ the partition function, cannot be analytically computed given the impossibility to compute the partition function when the system is highly dimensional. The Monte Carlo method consists in simulating a Markov process that fulfills detailed balance such that the limit distribution coincides with the Boltzmann distribution, in such a way that averages over the Boltzmann distribution can be substituted by averages over the Markov process.

This situation has a parallel in molecular evolution. In this case, our starting point is the transition probability of the Markov process (in the limit of a homogeneous population), from which we can easily compute the limit distribution in sequence space exploiting the detailed balance. *Sella & Hirsh (2005)* noted that this limit distribution has the form $P(A) = \frac{1}{Z}e^{N\log f(A)}$, i.e., sequences with higher fitness are visited more often during the course of evolution, as we expect, and the prevalence of sequences with large fitness is modulated with a probability that depends on population size. Strikingly, there is a strong formal analogy between this limit distribution in sequence space and the Boltzmann distribution in structure space. Firstly, there is a correspondence between fitness and minus energy where sequences with larger fitness are visited more often, in analogy with structures with lower energy that are observed more often. Secondly, there is another interesting correspondence between the effective population size and the inverse of the temperature, implying that small populations are more tolerant to sequences with low fitness in the same way in which systems with large temperature are more tolerant to structures with high energy. We think that this analogy can help the intuitive understanding of the basis

of statistical mechanics for biologists, and can transmit key evolutionary concepts to physicists.

This introduction motivates the following exercise, which consists in simulating the evolution of a homogeneous population with size $N$, subject to random mutations with a given bias and selection on protein stability. As target native structure, we choose one of the solutions of the snake cube. The random process that we simulate consists of four steps:

1. **Mutation** One of the positions of the sequence is randomly chosen and mutated from H to P or viceversa. We consider that mutation bias, i.e., the rate of mutation from H to P and from P to H may be different. This bias is parametrized with a parameter $p$ that expresses the ratio between the rate at which an amino acid of type H mutates to P and the rate at which P mutates to H (this parameter suffices since the total mutation rate only affects the time scale of the problem). We extract the mutant site in two steps. Firstly, we extract a random number to decide whether the mutation is from H to P (probability $P_{HP} = pn_H/(pn_H + (1-p)n_P)$, where $n_H$ is the number of positions occupied by a H) or viceversa. Then, we extract with uniform probability one of the $n_H$ sites if the mutation is from H to P, or one of the $n_P$ sites otherwise.

2. **Fitness evaluation** We compute the folding free energy of the mutated sequence at temperature $T$ with Eq. (10). The computation can be accelerated noting that only terms involving the mutated residue change. The fitness is then evaluated with Eq. (12).

3. **Selection** We compute the fixation probability, Eq. (11), and we extract a random number to decide whether the mutated sequence get fixed or is eliminated.

4. **Sampling** Finally, we take statistics of the relevant quantities (fitness, $\Delta G$). If the mutation is accepted we use the new sequence, otherwise we use the previous wild type sequence. It is of course important to remember to update the counts even if the sequence does not change.

Starting from a random sequence, stability and fitness tend to increase on average, although with large fluctuations. We are interested in the stationary properties at large time, when the limit distribution is reached and the memory of the starting random sequence is lost. We propose to use the following method to numerically estimate the stationary value of the fitness. We define the average fitness at time $t$, $F(t)$ as

$$F(t) = \frac{1}{t}\sum_{k=1}^{t} f(A(k)) . \tag{13}$$

We now assume that the average fitness $F$ tends to its stationary value exponentially,

$$F(t) \approx F(0) + (F_\infty - F(0))\left(1 - e^{-\frac{t}{\tau}}\right). \tag{14}$$

This assumption allows us to derive the parameters in which we are interested, $F_\infty$ (stationary fitness) and $\tau$ (time scale) through the linear fit of $y = F(t+1) - F(0)$ versus

$x = F(t) - F(0)$:

$$(F(t+1) - F(0)) = (F(t) - F(0))e^{-1/\tau} + (F_\infty - F(0))(1 - e^{-1/\tau}). \qquad (15)$$

The slope of the fit gives $\tau$ and the intercept gives $F_\infty$. This computation allows us to discuss the advantages of analytical methods such as linear fit with respect to other alternatives. An alternative method for computing $F_\infty$ consists in discarding the first $t_0$ steps of the simulation and using only the last steps to compute the average. However, it is difficult to be sure that the transient that we discard is large enough to guarantee convergence and not too large to reduce statistics. Moreover, we would miss the interesting information on the time scale $\tau$. The second possibility is to perform a nonlinear fit of Eq. (14). However, non-linear fits require an heuristic optimization that does not guarantee convergence to optimal parameters. The linear fit allows an analytic solution that is preferable, in particular for complex fits involving many parameters.

Armed with these tools, we investigate the dependence of fitness (hence, stability) on the properties of the evolving population: the physical temperature $T$ at which evolution takes place, the population size $N$ and the mutation bias $p$. We will finally test the influence of the native structure.

### Temperature

Figure 8 shows some typical simulations reaching a stationary value under the proposed evolutionary model. Each step represents one fixed mutation. The first evolutionary simulation that we propose has the objective to study the effect of temperature, which enters the definition of fitness Eq. (12) through the factor $\beta = 1/k_B T$. In Fig. 9 we compare results obtained with $T = 1$, $T = 10$ ($\beta = 0.1$) and $T = 100$ ($\beta = 0.01$). For the sake of comparison, we also show results for random sequences. Simulations were made with population size $N = 50$. We will analyse the effect of modifying population size in the next section.

When $T$ is low ($T \lesssim 1$) the fitness function Eq. (12) tends to a binary function of stability: $f \approx 1$ if $\Delta G < 0$ and $f \approx 0$ otherwise. This corresponds to a neutral fitness landscape in which all proteins that are sufficiently stable are selectively equivalent and all proteins that are unstable are strongly rejected. Even random sequences show a bimodal fitness distribution with peaks at $f = 0$ and $f = 1$, as expected for an almost neutral fitness landscape, and fitness close to one can be achieved even by random sequences. For evolved sequences the most likely value of $\Delta G$ is only slightly below zero, where the fitness is close to one and the entropy in sequence space is large (*Taverna & Goldstein, 2002*).

If $T$ increases ($\beta = 0.1$), the fitness becomes a smooth function of stability and it is more difficult to accept mutations that decrease stability. As a consequence, the mean value is significantly lower than for $\beta = 1$ ($-12.64 \pm 0.04$ versus $-6.48 \pm 0.04$ for $\beta = 1$). For the same temperature, random sequences have free energies that are on average zero and, correspondingly, fitness distributed around $f = 0.5$, which is the value attained when $\Delta G = 0$. Under this point of view, the inverse of the physical temperature $1/T$ can be regarded as an evolutionary temperature, in that, the larger is $1/T$, the more tolerant is the evolution with respect to mutations that decrease protein stability.

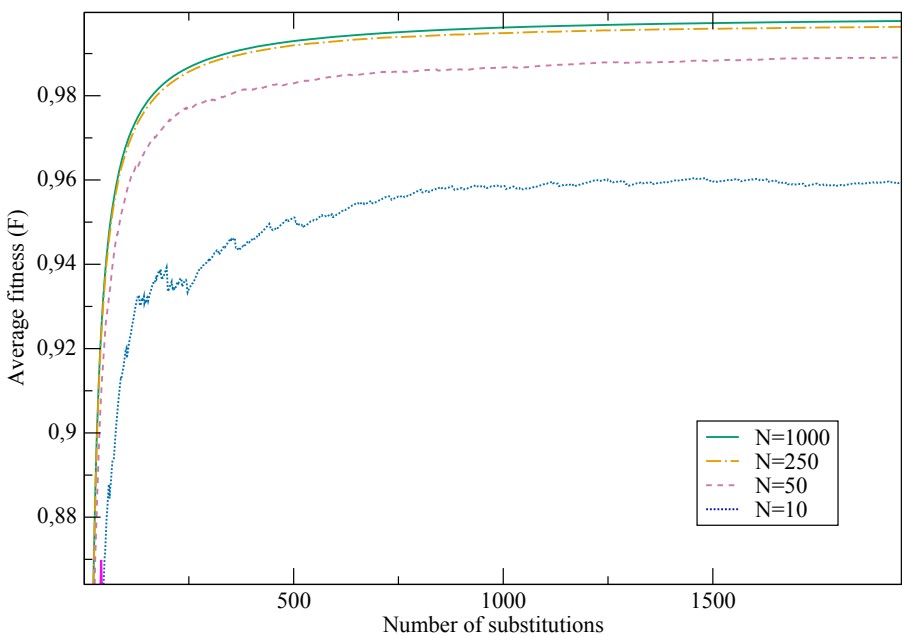

**Figure 8  Average fitness versus the number of fixed mutations.** Each curve represents an evolutionary trajectory for a given population size.

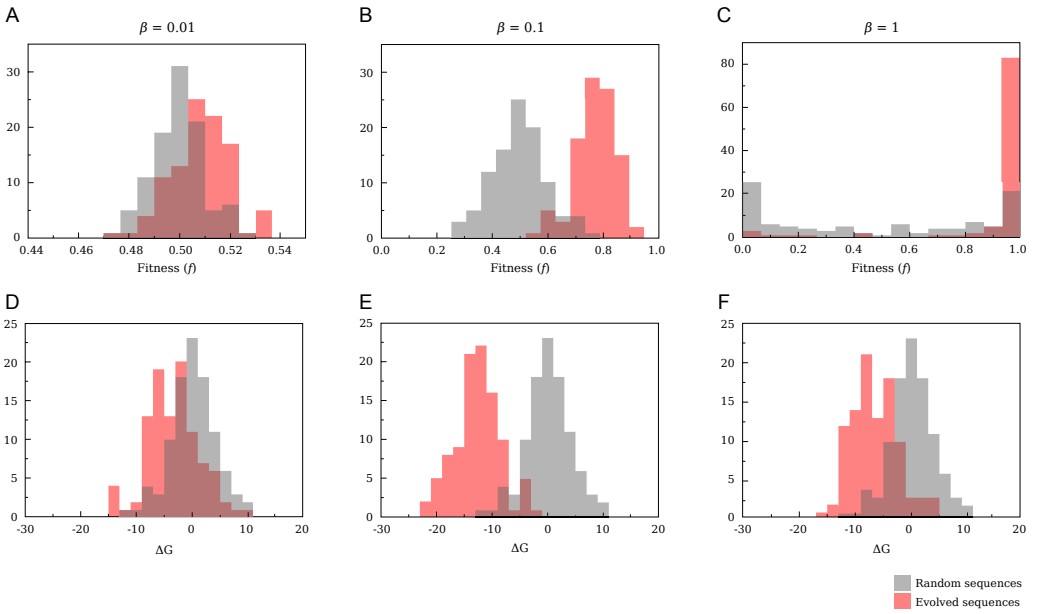

**Figure 9  Comparison of the distributions of fitness (A–C) and free energy (D–F) for randomly drawn and evolved sequences.** The evolutionary parameters are $N = 10$; $p = 0.5$. For $T = 1$ (C,D) we obtain a neutral landscape, and the variation of free energy after the evolutionary process is smaller than for $T = 10$ (B,E). Increasing temperature until $T = 100$ (A,D) all sequences have almost the same fitness, and evolution becomes an almost random process.

In the high temperature limit ($\beta = 0.01$) all sequences have the same fitness $f \approx 0.5$ independent of $\Delta G$ and evolution becomes an almost random process. Indeed, it is difficult to obtain higher values for evolving sequences and the mean value of the free energy for evolved sequences, although still negative, approaches zero—with a mean value of $-3.24 \pm 0.5$.

## Neutrality

An important consequence of the functional form of the fixation probability Eq. (11) is that the closer $F$ is to the neutral regime in which fitness is a step function of stability, the less the evolutionary process depends on population size. In particular, as we have seen above, for low temperature protein stability approaches the neutral threshold (*Taverna & Goldstein, 2002*), while in a non-neutral regime the equilibrium stability strongly increases with population size. For neutral evolution the substitution rate, i.e., the rate at which mutations become fixed in the population, is independent of $N$. In fact, if $\mu$ denotes the mutation rate and $x$ denotes the probability that mutations are neutral, there will be on average $N\mu x$ neutral mutations arising in the population per unit time. By Eq. (11), when the fitness of the wild-type and the mutant are equal the probability of fixation tends to 1/N, and the average number of fixations per unit time tends to the neutral mutation rate $\mu x$, independent of population size, as predicted by Kimura's neutral model (*Kimura, 1980*).

## Population size

The next exercise proposes to study the effect of the effective population size $N$. This may be a good opportunity to discuss the concept, explaining that the effective population size is not just the number of individuals in the population but it is a number that recapitulates its demographic history, in particular bottlenecks. We plot in Fig. 10A the mean value of the free energy in the stationary state (See Supplementary Methods). When the temperature is low ($T \lesssim 1$) the outcome of the evolutionary process is almost independent of the population size $N$, a hallmark of neutral evolution. When $T$ increases and the fitness is a smooth function of stability, the outcome of evolution strongly depends on population size, in the sense that the equilibrium stability markedly increase with population. Finally, for high temperature (e.g., $\beta = 0.001$), evolved sequences have almost identical properties to those of random sequences except for very large population size ($N\beta \approx 1$).

We show in Fig. 10B the effect of changing temperature for fixed population sizes. As it was shown in the previous section for fixed population size ($N = 50$), a non-neutral regime is observed for intermediate temperature. This effect is more pronounced for larger population sizes, and the free energy reaches a minimum close to $\beta = 0.1$ for most values of $N$. We conclude this exercise showing in Table 2 the equilibrium values of fitness and stability obtained for different population sizes keeping all the other parameters fixed, within a non-neutral temperature regime ($\beta = 0.5$). We chose this value in such a way that free energy values are clearly distinguishable between any pair of population sizes (see Fig. 10). We can see that smaller populations attain significantly smaller equilibrium fitness, since the fixation probability of disadvantageous mutations is larger for smaller

A

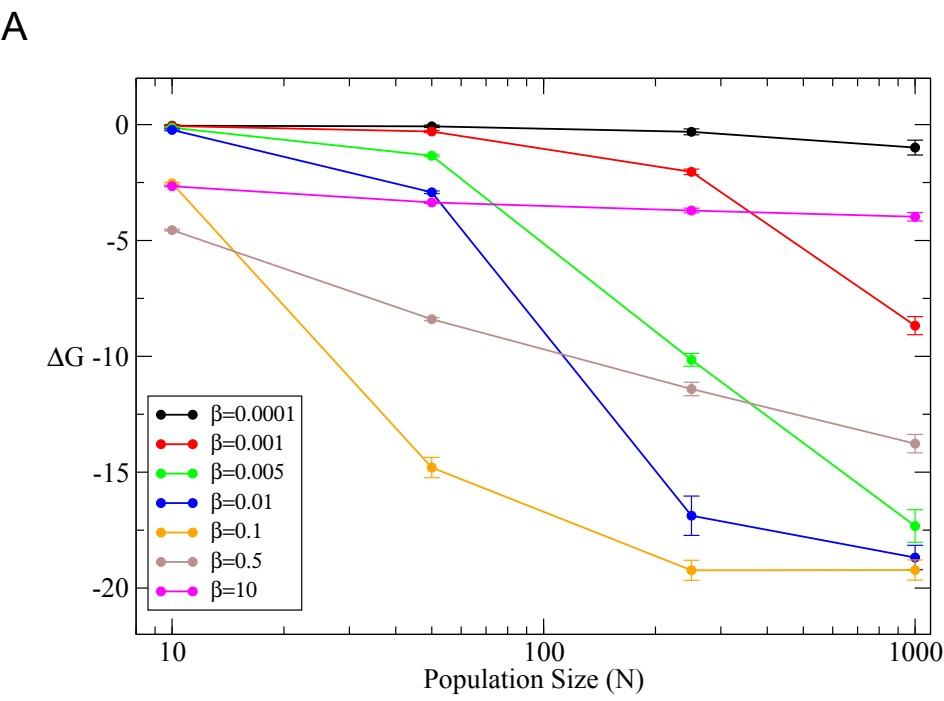

B

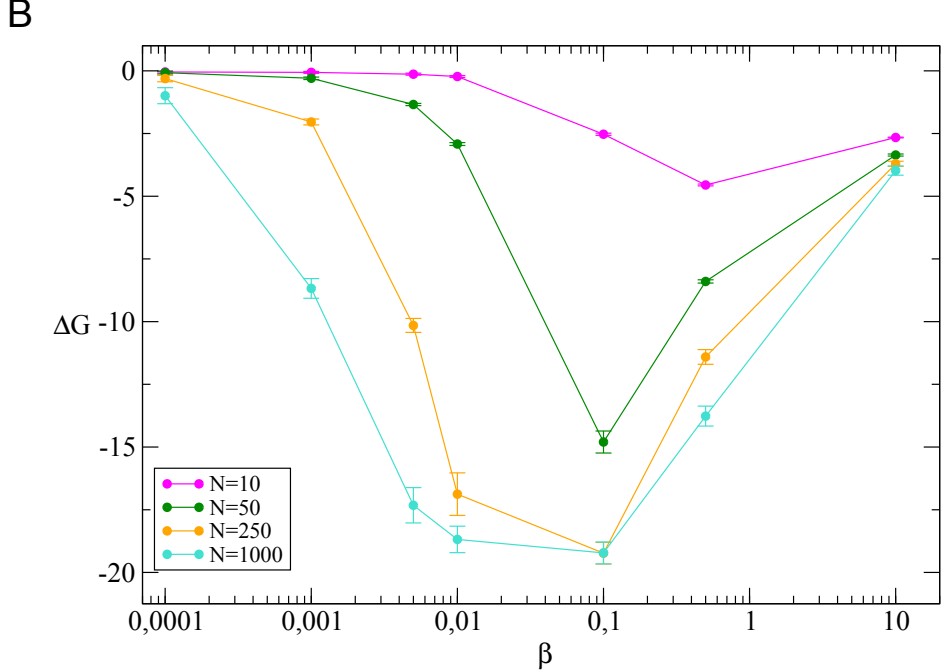

**Figure 10** **Average free energies of evolved sequences.** (A) Versus population size for different temperatures ($\beta = 1/k_{\mathrm{B}}T$). (B) Versus temperature for different population sizes.

**Table 2  Average and standard deviation for the fitted parameters and free energy obtained in 10 independent evolutionary processes.** The parameters are $p = 0.5$ (balanced mutations) and inverse temperature $\beta = 0.5$.

|  | $N = 10$ | $N = 50$ | $N = 250$ | $N = 1,000$ |
|---|---|---|---|---|
| $F_{\infty}$ | $0.8682 \pm 0.0007$ | $0.9770 \pm 0.0001$ | $0.9948 \pm 0.00015$ | $0.9984 \pm 0.00015$ |
| $\tau$ | $154 \pm 39$ | $200 \pm 42$ | $267 \pm 63$ | $308 \pm 77$ |
| $\Delta G$ | $-4.552 \pm 0.0036$ | $-8.397 \pm 0.065$ | $-11.42 \pm 0.29$ | $-13.76 \pm 0.39$ |

**Table 3  Averages and standard deviations for the fitted parameters and free energy obtained in 10 independent evolutionary simulations.** The parameters used are $N = 50$ and $\beta = 0.5$. The target structure used is S7. Different mutation bias are represented.

|  | $p = 0.25$ | $p = 0.5$ | $p = 0.75$ |
|---|---|---|---|
| $F_{\infty}$ | $0.9764 \pm 0.0002$ | $0.9770 \pm 0.0001$ | $0.9764 \pm 0.0002$ |
| $\tau$ | $247 \pm 66$ | $200 \pm 42$ | $250 \pm 60$ |
| $\Delta G$ | $-8.201 \pm 0.047$ | $-8.397 \pm 0.065$ | $-8.120 \pm 0.045$ |

population, in agreement with the statistical mechanical analogy (*Sella & Hirsh, 2005*). Moreover, larger populations also attain significantly larger stability, which shows that the system is far from the neutral regime in which fitness is a stepwise function of stability.

For these simulations we observe an exponential trend towards the stationary state and, therefore, we can compute the asymptotic time $\tau$ through a linear fit, as explained above. Nevertheless, the differences in time scales between the different populations, although systematic, are not significant. In fact, $\tau$ depends on the initial value of the fitness, which is a variable that fluctuates strongly, so that a very large number of independent simulations would be necessary to reduce the variability and improve the significance.

### Mutation bias

Finally, we propose to explore the role of the mutation bias $p$ (see Materials and Methods). We perform these simulations for population size $N = 10$ and high temperature $\beta = 0.1$, since $p$ is expected to have a more relevant impact for small populations and non-neutral fitness landscapes. Table 3 displays results for different values of the mutation bias $p = 0.25,\ 0.5,\ 0.75$ corresponding to hydrophobic, neutral and polar sequences, respectively. We see that the equilibrium values of the free energy $\Delta G$ display a significant change. As shown in Table 3, the stability against unfolding (i.e., the free energy of the native state) decreases when the mutation bias favours polar sequences and the stability against misfolding (i.e., the free energy of misfolded conformations) has the opposite behaviour. This data also suggest that the folding free energy resulting from both unfolding and misfolding is minimum when the mutation bias is close to the mutation bias $p = 0.5$ at which polar and hydrophobic mutations balance, at least for the chosen parameters (see the quadratic fit in Fig. 11, which has illustrative purposes only).

The study of the mutation bias raises two important points from the perspectives of physics and evolution. From the point of view of physics, it allows us to note an important difference between the Monte Carlo process used to simulate the Boltzmann distribution

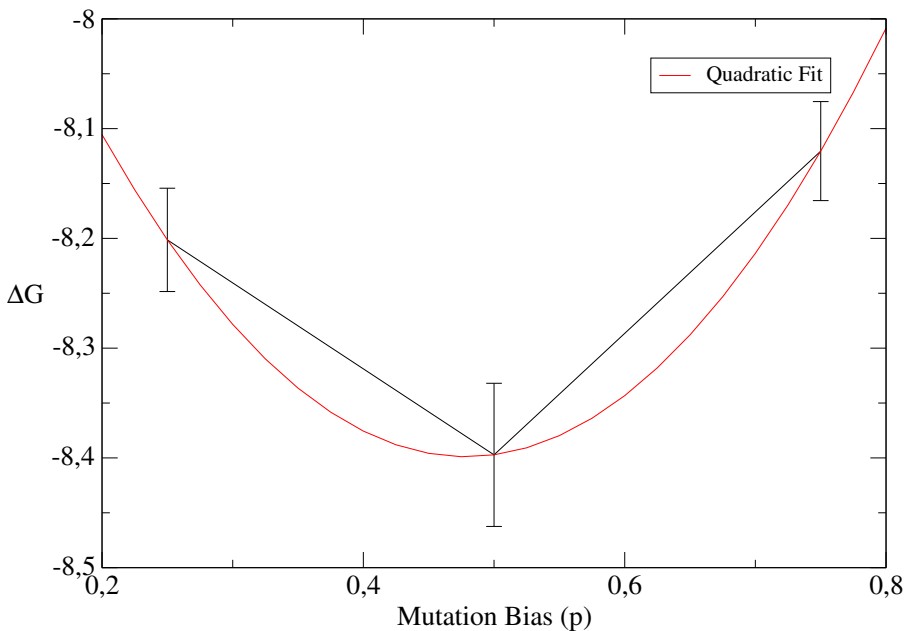

**Figure 11 Dependence of the free energy on the mutation bias.** A quadratic fit is shown for illustrative purposes only.

in statistical physics and the evolutionary process. While in the case of statistical physics the equilibrium distribution does not depend on which mutations are proposed, and it always coincides with the Boltzmann distribution by construction, in evolution the limit distribution does depend on the mutation process, and even key properties such as the equilibrium fitness and the stability of proteins are influenced by the mutation bias, as it was shown in *Méndez et al. (2010)*.

This fact reflects an important difference between the equilibrium distribution in the phase space of statistical physics, i.e., the Boltzmann distribution, and the equilibrium distribution in the space of evolving protein sequences. While the Boltzmann distribution can be defined as the probability distribution with maximum entropy given a constraint on the average energy, the equilibrium distribution in sequence space can be defined as the distribution with minimum Kullback–Leibler divergence from the distribution that would be attained by mutation alone, given the constraint on average fitness (*Arenas, Sánchez-Cobos & Bastolla, 2015*). The correspondence between the two definitions can be appreciated noting that the entropy is equal to the Kullback–Leibler divergence from the equiprobable distribution, i.e., in the case of statistical physics the reference distribution is the equiprobable distribution, while in the case of evolution the reference distribution is the distribution that would be attained by mutation alone.

From the point of view of evolution, the fact that the equilibrium fitness depends on the mutation process creates the possibility of an interesting feedback between selection and mutation, which in turn depends on the replicative machinery of the organism and is under genetic control. In other words, mutation and selection should not be considered as completely independent processes, but there is the possibility that a population selects

**Table 4  Averages and standard deviations for the fitted parameters and free energies obtained in 10 independent evolutionary simulations.** The parameters considered are $N = 50$ and $\beta = 0.5$ using the structures S7 (high designability) and S1 (low designability) as target. See 'Methods' for further details.

| structure | S1 | S7 |
|---|---|---|
| $F_\infty$ | $0.9766 \pm 0.00015$ | $0.9770 \pm 0.0001$ |
| $\tau$ | $232 \pm 47$ | $200 \pm 42$ |
| $\Delta G$ | $-8.386 \pm 0.052$ | $-8.397 \pm 0.065$ |

the mutation process that is more convenient to it under its ecological and evolutionary circumstances (*Méndez et al., 2010*).

### Structural effects

Finally, we explore the influence of the target structure in the evolutionary process by comparing the least designable structure S1 and the most designable structure, S7 (see Table 4). The structure S7 reaches higher free energy on average, but with a much smaller value of the time scale $\tau$, which indicates that it reaches equilibrium faster, consistent with the fact that there are more sequences for which this structure is lower in energy.

## Sequence-structure relationship

We conclude our analysis by relating the statistical properties of the evolved sequences and their corresponding structures. This is particularly simple in the HP model, in which each position along the sequence can be characterized by a single parameter representing the frequency at which the position is occupied by a hydrophobic residue in the given ensemble of sequences. For real proteins, each position must be characterized by a 19 dimension vector of the frequency of the different amino acid types (the 20-th frequency is the result of the normalization condition), called *profile* in the bioinformatics literature. We plot in Fig. 12 the hydrophobicity profiles of random sequences and sequences evolved to fold into structure S7 for mutation bias $p = 0.5$ (see Materials and Methods for further details). The profiles of random sequences only depend on the mutation bias but, by definition, they do not differ significantly between one position and the other. In contrast, we observe marked differences between positions for the profiles of evolved sequences. To rationalize these differences, we also report in Fig. 12 the number of contacts at each position of the target structure. The predictive correlation between this structural profile and the sequence profile is readily apparent, and implies that in this model the number of contacts determines the hydrophobicity profile of each position (*Bastolla et al., 2008*). Since positions with many contacts are buried in the interior of the native structure, the model recovers the well-known property of real proteins that buried positions tend to be hydrophobic and surface positions tend to be polar, and it shows that this tendency is sufficient to completely determine the hydrophobicity profile in the simple case of the HP model.

It can be interesting to suggest further readings that show that the contact energy model Eq. (10) together with a statistical mechanics approach allows us to analytically predict the observed correlation between the number of contacts and the hydrophobicity profile (*Porto et al., 2005*; *Arenas, Sánchez-Cobos & Bastolla, 2015*). This point can be proposed as

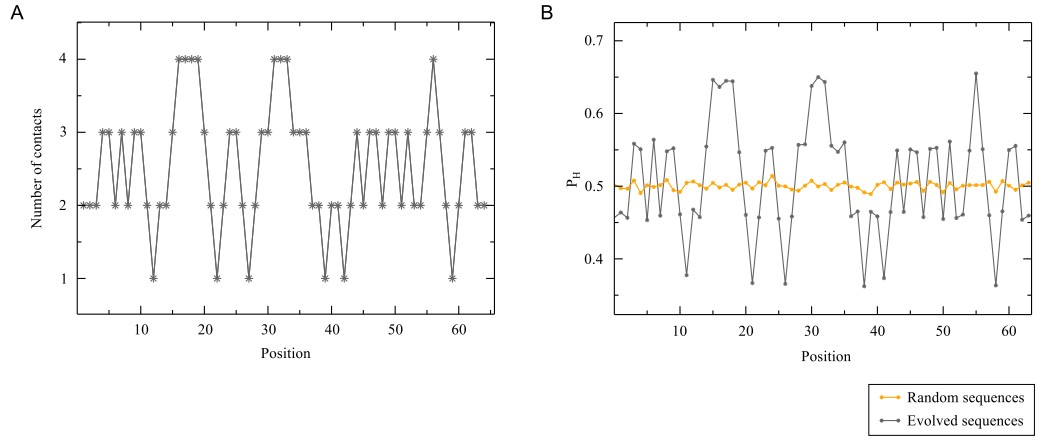

**Figure 12 Sequence and structure profiles.** (A) Number of contacts for each position of structure S7. (B) Frequency of hydrophobic residues at a given position for random sequences and sequences evolved to fold into S7. We can see that the number of contacts is a reliable predictor of the frequency of hydrophobic residues of evolved sequences. This is also consistent with the common observation that deeply buried positions, which have a larger number of contacts, tend to be more hydrophobic.

an exercise, and it can be an interesting way to introduce the Boltzmann distribution in statistical physics and to deepen the analogy between statistical mechanics and evolution.

One consequence of the structural propensities discussed above is that the positions of evolved sequences are more conserved than positions in the random ensemble. We can quantify this through the sequence entropy of each position $i$, defined as $S_i = -\sum_a p_i(a) \ln(p_i(a))$ where $a$ denotes one of the two amino acid types and $p_i(a)$ is the profile at position $i$. The sequence entropy is smaller for evolved than for random sequences. This reduction of entropy is the consequence of the constraints that enforce the stability of the native structure.

## FINAL REMARKS

In this paper we presented a teaching unit dedicated to the computational study of protein structures, stability and evolution. This area of research is difficult to present to students, since it requires a highly multidisciplinary background that includes statistical mechanics, evolution and computational skills. Our experience teaching this subject to students at the Master's program of Biophysics of the Universidad Autónoma of Madrid (Spain) has shown us that these concepts are easier to present using as a toy model a real toy such as the snake puzzle. This allowed us making abstract concepts more concrete, and opened several avenues towards more advanced subjects of evolutionary and structural bioinformatics. We have selected exercises that, in our opinion, establish simple parallelisms with interesting and simple enough publications that we suggest to the students for further reading.

We want to remark that these exercises are not intended to replace other educational tasks that deal with real biological entities. Rather, we believe that the proposed approach aids students to establish a fundamental theoretical framework through a computationally and conceptually tractable model. Ultimately, these exercises are to provide a foundation

upon which students can build increasingly complex biophysical models and design strategies to tackle real biological problems.

In our teaching experience, we have realized that students that lack a background in physics face, in general, the most difficult conceptual challenges. For these students, the evolutionary model that we propose may constitute an intuitive introduction to statistical mechanics concepts, and the simulations proposed constitute a practical introduction to scientific computation. In general, we find that students quickly build an intuition on the problem. On the other hand, students with a background in physics usually enjoy this new application of statistical mechanics, but they tend to have severe difficulties to interpret the results from a biological point of view. In our experience, a fruitful way to take advantage of these differences consists in forming working teams of students with different backgrounds.

In summary, the increasingly interdisciplinary setting of scientific research requires efforts to overcome traditional boundaries between established academic disciplines. These efforts are yielding promising results through the application of interesting alternatives (*Searls, 2012*). A suitable scientific academic curriculum should also be evaluated by assessing how students are getting on with their early scientific career stages. In this regard, we believe that incorporating interdisciplinary lectures in the design of academic curricula is of key importance, not only for computational biology (*Fox & Ouellette, 2013*). It is our responsibility to claim for these changes.

## MATERIALS AND METHODS

### Algorithm to solve the snake puzzle

To solve the snake puzzle, we adopted a straightforward algorithm that builds the conformations of the snake iteratively and implements an exhaustive search of all maximally compact conformations, i.e., conformations that can be fitted into a cube of side 3 (for the 27-*mer*) or 4 (for the 64-*mer*). The search is performed on a decision tree whose nodes correspond to spatial arrangements of the consecutive rigid fragments. From each node, there are four possible directions where the next fragment can be placed, since two consecutive fragments cannot be extended in the same direction. For example, if fragment $i$ is placed along the $x$-axis, fragment $i+1$ can be placed only in the $+/-y$ or $+/-z$ directions. The first two rigid fragments are placed together at the root of the tree and define the oriented $x$ and $y$-axes. The third fragment can be placed in any direction perpendicular to $y$, i.e., $+x$, $-x$, $+z$ and $-z$. The last two options are related by mirror symmetry (see also Main Text), which we can reduce allowing only the placement in the $+z$ direction the first time that the $z$ axis is visited.

At each step, the algorithm tests whether the new fragment occupies positions already occupied by other fragments (self-avoidance condition) and whether the partial conformation extends outside the boundaries of the cube. If any of these requirements is not fulfilled, the node is discarded and the algorithm goes back to the parent node, moving in the remaining directions. Otherwise, we create the node corresponding to the new accepted conformations and proceed forward. When all four directions have been tested, the algorithm goes back to the parent node. Note that the tree can be built starting

with any fragment of the puzzle, in particular the initial and final fragments, but also some in between (in this case, it has the choice to start moving forward and then backward or vice versa).

We represent the final solution as an ordered string containing, for every fragment, its direction with sign, e.g., $(+x, +y, -x, \ldots, +z)$. This format can be converted to explicit coordinates that can be input to the visualization software used in structural bioinformatics (see Main Text) to visually inspect the solutions, and used to solve the puzzle manually.

## Designability and energy gap

To compute the designability of each of the non-redundant solutions (we exclude S2 since it is very similar to S1, see Main Text), we randomly draw $m$ sequences of the HP model (i.e., only two amino acid types H and P are considered) with probability $p = 0.5$ that the amino acid is P. For each sequence we compute the effective energies of the target structures using Eq. (2), with the same parameters used by *Li et al. (1996)* ($U(H, H) = -2.3$, $U(H, P) = -1$ and $U(P, P) = 0$) and assign the sequence to the structure with lowest energy. In this way, we compute the fraction of sequences assigned to each structure. The experiment is replicated 100 times to evaluate the statistical error, with set sizes $m = 10, 100, 1,000$ and 10,000. The average and standard error of the mean are plotted as a function of $m$ in Fig. 6.

## Structurally constrained sequence evolution

We simulate protein sequence evolution with structural constraints using a Monte Carlo algorithm illustrated in the Main Text. We extract the initial sequence $A(0)$ of length $L = 64$ and compute its free energy Eq. (10) and its fitness, Eq. (12). At each step $t$ of the simulation we mutate a random position of the sequence with bias $p$ for polar replacement. We then compute the free energy and fitness of the new sequence and obtain the probability of fixation $P_{fix}$, (11). We then extract a random number $0 \leq r \leq 1$ and we accept the new sequence (i.e., $A(t+1) = A'$) if $r < P_{fix}$. Otherwise, the old sequence is kept ($A(t+1) = A(t)$). It is important to note that, when the old sequence is kept, its associated evolutionary values are recorded one more time. Considering values associated to rejected mutations is needed to ensure that the underlying distribution sampled within the stationary state is indeed a Boltzmann distribution, from which we next compute the average of the fitness and free energies.

We perform simulations changing some key parameters: nine different temperature values distributed between $\beta = 0.0001$ to $\beta = 10$ (see Fig. 10), three different values for the mutation bias ($p = 0.25, 0.5\ 0.75$) that the mutation is from a hydrophobic to a polar amino acid, four different population sizes ($N = 10, 50, 250, 1,000$), and two different target structures (S1 and S7). The combinations of parameters selected for the simulations are described in the main text. For each set of parameters, 10 independent simulations were run until the stationary was reached.

From the fitness values recorded, the average fitness $F$ is computed, and the stationary fitness $F_\infty$ and the evolutionary time scale $\tau$ obtained through the linear fit described in the main text, Eq. (15). The average of the free energy $\Delta G$ and its standard error were computed considering a sufficiently large number of points within the stationary regime.

The values presented in the tables and figures correspond to the mean of the 10 averaged values, and the errors to the propagation of the average errors obtained from the 10 runs.

## ACKNOWLEDGEMENTS

We acknowledge Pablo Mateos-Gil for bringing to our attention the snake puzzle as a toy model of protein structure. We are particularly in debt with Raúl Guantes, coordinator of the former Master on Biophysics of the Universidad Autónoma de Madrid (now Master on Condensed Matter Physics and Biological Systems), for his support in the development of these lectures, and to the students for their motivation and feedback.

### Funding

This work is supported by the Marie Curie Training Network NETADIS (FP7, grant 290038) (LBR), by the Comunidad de Madrid (Amarauto program to UB), by the Spanish Ministry of Economy and Competitiveness (FPI grant BES-2009-013072 to APG and BFU2012-40020 to UB). Research at the CBMSO is facilitated by the Fundación Ramón Areces. The funders had no role in study design, data collection and analysis, decision to publish, or preparation of the manuscript.

### Grant Disclosures

The following grant information was disclosed by the authors:
Marie Curie Training Network NETADIS: 290038.
Comunidad de Madrid.
Spanish Ministry of Economy and Competitiveness: BES-2009-013072, BFU2012-40020.
Fundación Ramón Areces.

### Competing Interests

Ugo Bastolla is an Academic Editor for PeerJ.

### Author Contributions

- Gonzalo S. Nido and Ludovica Bachschmid-Romano performed the experiments, analyzed the data, wrote the paper, prepared figures and/or tables, performed the computation work, reviewed drafts of the paper.
- Ugo Bastolla conceived and designed the experiments, analyzed the data, wrote the paper, reviewed drafts of the paper.
- Alberto Pascual-García conceived and designed the experiments, analyzed the data, wrote the paper, prepared figures and/or tables, performed the computation work, reviewed drafts of the paper.

## Data Availability

The sequence of the rigid elements of the snake cube puzzle, the contact matrices and the script used to solve the puzzle can be found in the URL https://github.com/insectopalo/snake-puzzle

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
