# Peer review of "Learning structural bioinformatics and evolution with a snake puzzle"

_PeerJ Computer Science, doi:10.7717/peerj-cs.100_

## Round 0.1 · original submission · Minor Revisions

The paper, although not a traditional scientific contribution, has a significant importance for teaching. Several of reviewers liked the submission on that basis. If the comments from second reviewer can be addressed in some way, that would be great.

Reviewer 1 ·

Basic reporting

The content of this paper is well organized and the text is clearly written. However there are some typos, which could benefit from a proofread. I have only two minor comments about the figures:

Figure1 could be dissected into 3 parts and labeled as "a","b","c". a) maximally compact conformation, b)intermediate conformation resembling a protein hinge, c) linearly arranged conformation resembling intrinsically disordered regions.

Figure 2 needs y axis label.

Experimental design

The authors are describing a methodology for teaching graduate students "protein folding and evolution" with the help of Snake Puzzle analogy, which they have been using in their Master course at Universidad Autonoma de Madrid.

They provide sufficient information to reproduce their method.

Validity of the findings

The proposed game/method to enhance the teaching practice is sound and robust. I believe that the students coming from different backgrounds will benefit from this paper.

Reviewer 2 ·

Basic reporting

The paper designed a snake puzzle to teach students the protein folding problem. The puzzle is similar to the e lattice models of protein structures. The puzzle is interesting. However, many parts of this paper are just reintroduction of the existing knowledge, such as RMSD, structure alignment, protein structure classification, etc.

Experimental design

From the abstract and introduction, the authors claims that to design teaching basic concepts of protein folding with the snake puzzle. However, the major of the paper is presenting the common concepts in the protein folding and relevant area. The experimental design is loosely related to the topic claimed.

Validity of the findings

no comments.

Additional comments

There are three problems that the authors might want to address in this paper. First, the authors may want to design the a puzzle to help teaching protein folding. Second, the authors may want to solve the puzzle. Third, the authors may want to solve the protein folding problem under lattice model. In any case, the authors should state clearly which problems to be addressed, and how the paragraphs are related to these problems. In addition, the commonly known paragraphs can be shorten, and references can be added.

·

Basic reporting

This paper proposes a working unit to teach a course about protein folding and evolution. The authors proposes to use a wooden snake puzzle as a starter, where various concepts on combinatorics, modeling, folding, alignment are introduced, and related software in the literature can then be applied to solve a problem. After that,
other concepts like evolution model and related statistical analysis are developed.

This working unit has been used in teaching a master course with students of inter-disciplinary background.

Experimental design

No comments.

Validity of the findings

The description of the working unit is very detailed, and interesting.
I believe this working unit is suitable for a master course as suggested.
The only thing I am concerned is how to test the relative effectiveness of using this working unit, as opposed to other ways (such as: teaching the same topics, but working directly with real data instead of the snake puzzle).

Additional comments

The paper is generally well-written.
There are some potential very grammatical mistakes that the authors may wish to go through the whole manuscript carefully, such as:

Line 50: "... that help bridging the gap ..." --> "... that help bridge the gap ..." ?

---

## Round 0.2 · accepted · Accept

We appreciate your efforts in clarifying the scope and contribution of these project in terms of teaching. Thank you for the clarifications provided.